# Archaean zircons in Miocene oceanic hotspot rocks establish ancient continental crust beneath Mauritius

Lewis D. Ashwal[1], Michael Wiedenbeck[1,2] & Trond H. Torsvik[1,2,3,4,†]

A fragment of continental crust has been postulated to underlie the young plume-related lavas of the Indian Ocean island of Mauritius based on the recovery of Proterozoic zircons from basaltic beach sands. Here we document the first U–Pb zircon ages recovered directly from 5.7 Ma Mauritian trachytic rocks. We identified concordant Archaean xenocrystic zircons ranging in age between 2.5 and 3.0 Ga within a trachyte plug that crosscuts Older Series plume-related basalts of Mauritius. Our results demonstrate the existence of ancient continental crust beneath Mauritius; based on the entire spectrum of U–Pb ages for old Mauritian zircons, we demonstrate that this ancient crust is of central-east Madagascar affinity, which is presently located ∼700 km west of Mauritius. This makes possible a detailed reconstruction of Mauritius and other Mauritian continental fragments, which once formed part of the ancient nucleus of Madagascar and southern India.

[1] School of Geosciences, University of the Witwatersrand, Private Bag 3, WITS, Johannesburg 2050, South Africa. [2] Deutsches GeoForschungsZentrum GFZ, Telegrafenberg, D14473 Potsdam, Germany. [3] Center for Earth Evolution and Dynamics (CEED), University of Oslo, 0316 Oslo, Norway. [4] Geodynamics, Norges Geologiske Undersøkelse (NGU), N-7491 Trondheim, Norway. † Present address: Deutsches GeoForschungsZentrum GFZ, Telegrafenberg, D14473 Potsdam, Germany. Correspondence and requests for materials should be addressed to L.D.A. (email: lewis.ashwal@wits.ac.za).

Evidence is accumulating that old continental crust may occur beneath some young ocean-island volcanoes, contributing contaminating components to their chemical and isotopic compositions. For example, the Sr, Pb and Nd isotopic compositions of basalts to rhyolites in the Öræfajökull volcano of southeastern Iceland were modelled to have incorporated 2–6% of Precambrian continental crust[1]. This, combined with inversion of gravity anomaly data that indicate unusually thick crust of >30 km, has been used to infer that the Jan Mayen Microcontinent extends southwestward to underlie southeast Iceland[1]. Similarly, a fragment of continental crust was suggested to underlie the young plume-related lavas of the Indian Ocean island of Mauritius, on the basis of gravity inversion modelling (crustal thickness) and the recovery of Proterozoic (660–1,971 Ma) zircons from basaltic beach sands[2]. The island of

Mauritius (2,040 km[2]) is the second youngest member of a hotspot track extending from the active plume site of Réunion, through the Mascarene Plateau, the Laccadive-Chagos Ridge and into the 65.5 Ma Deccan Large Igneous Province[3,4]. Three phases of Mauritian basaltic volcanism have been named the Older (9.0–4.7 Ma), Intermediate (3.5–1.66 Ma) and Younger (1.0–0.03 Ma) Series[5,6]. Minor volumes of trachytic rocks occur as intrusive masses or plugs <300 m across and are associated with Older Series basalts, as confirmed by U–Pb zircon thermal ionization mass spectrometry dating, which yielded at one location an age of 6.849 ± 0.012 Ma (ref. 7). Mauritian trachytes are variably altered from probable primary phonolitic magmas[7], forming a prominent Daly Gap when plotted with coeval basalts; this suggests formation either by extreme fractional crystallization from the basalts[8] or by direct partial melting of

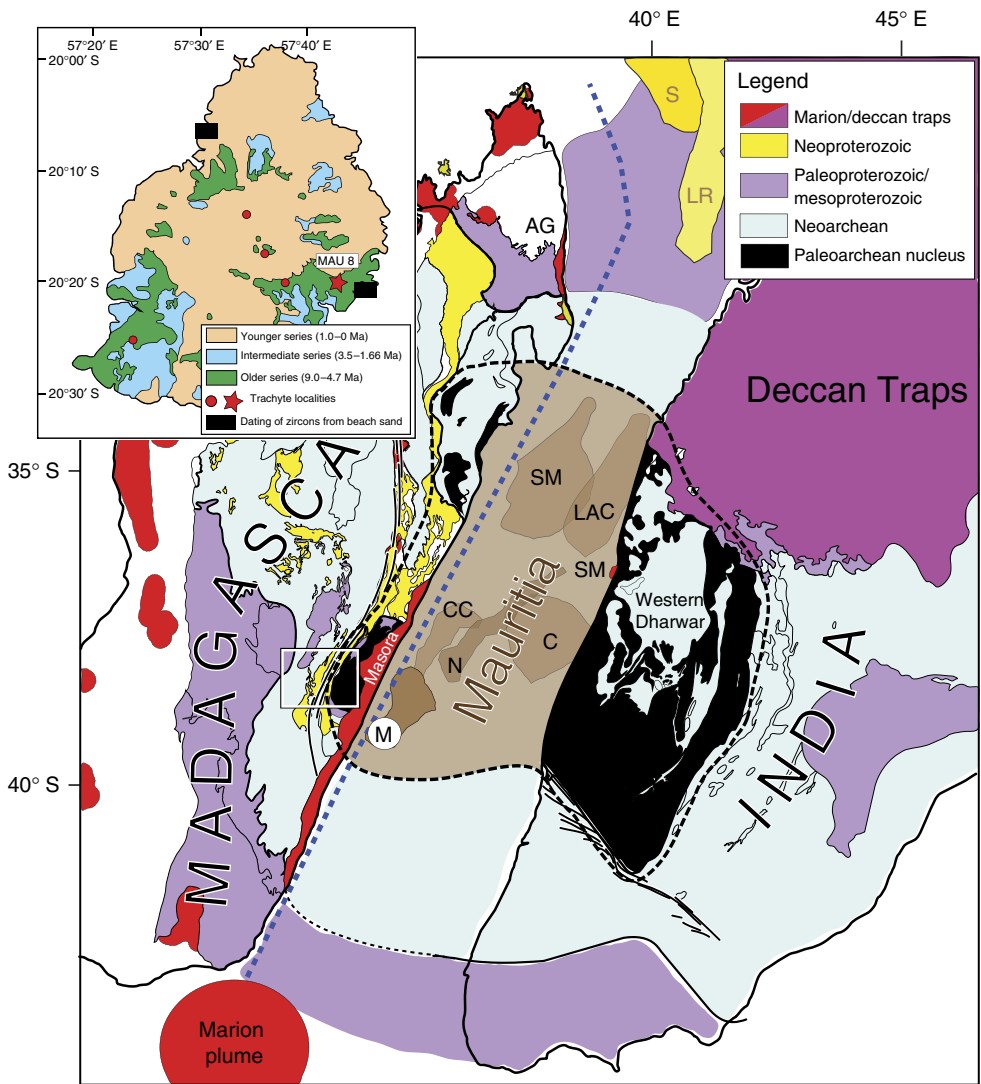

**Figure 1 | Simplified geology of Madagascar and India reconstructed to 90–85 Ma.** Mauritius (M) is reconstructed in a likely location near Archaean–Neoproterozoic rocks in central-east Madagascar just prior to break-up[2]. The exact size and geometries of Mauritius and other potential Mauritian continental fragments (collectively known as Mauritia, including SM Saya de Malha; C, Chagos; CC, Cargados-Carajos Banks; LAC, Laccadives; N, Nazreth; see present location in Fig. 6) are unknown, and are generously drawn in the diagram. We propose that Mauritia is dominantly underlain by Archaean continental crust, and part of the ancient nucleus of Madagascar[25,46] and India[20,21] (stippled black line). A Large Igneous Province event (linked to the Marion plume) occurred from 92 to 84 Ma, and most of Madagascar was covered with flood basalts (full extent not shown for simplicity). Blue stippled line indicates the site of Cretaceous pre-breakup strike-slip faulting. AG, Analava gabbro (91.6 Ma); LR, Laxmi Ridge; S, Seychelles; SM, St Mary rhyolites (91.2 Ma)[41]. The black–white box (geology of Madagascar) is enlarged in the inset to Fig. 5. Inset map shows simplified geology of Mauritius, including trachyte plugs[7]. Star symbol marked MAU-8 is the sampling area for the present study and black bars indicate locations of zircons recovered from beach sand samples[2].

metasomatized mantle[7]. Isotopic compositions of the trachytes show relatively constant $\varepsilon_{Nd}$ of $+4.03\pm0.15$, with highly variable $I_{Sr}$ of 0.70408–0.71034, interpreted as reflecting small amounts (0.4–3.5%) of contamination by Precambrian continental crust, successfully modelled using as a proxy $\sim$750 Ma granitoid composition from the Seychelles. It has been noted that other compositions, including older Proterozoic or even Archaean components, can also be plausibly modelled[7]. It is from such a

Mauritian trachyte that a population of zircons was extracted and analyzed for this paper. A subset of these zircons yields Archaean U–Pb ages, confirming the existence of a fragment of Precambrian continental crust beneath Mauritius, and we propose here that Mauritius and other Mauritian continental fragments are dominantly underlain by Archaean continental crust, and that these originally formed part of the ancient nucleus of Madagascar and India.

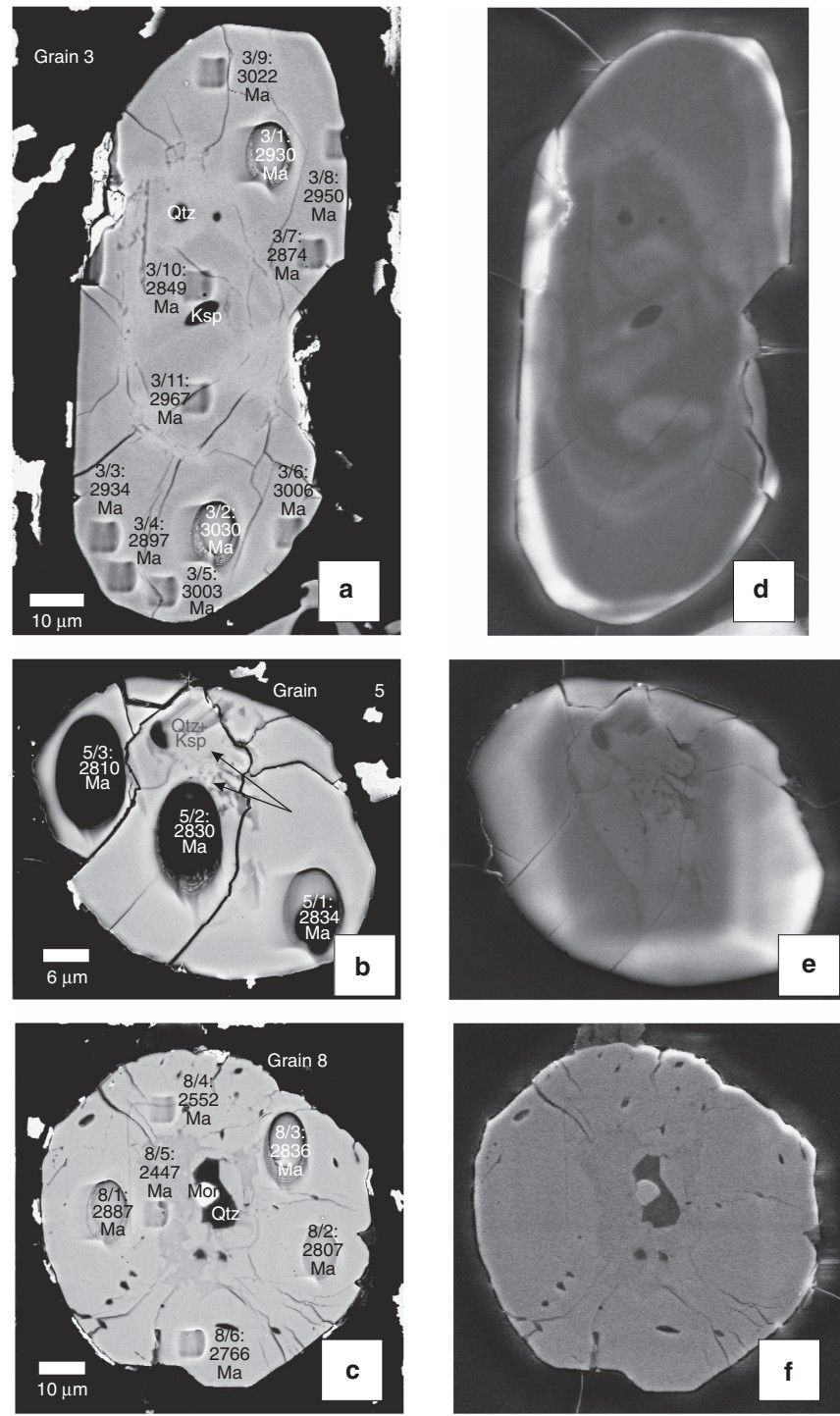

**Figure 2 | Scanning electron microscope images of three Archaean zircon grains.** These three grains were recovered from the MAU-8 trachyte sample. Backscattered electron (BSE) images (**a–c**) of the three grains taken after completing all U–Th–Pb isotopic analyses. Cathodoluminescence (CL) images (**d–f**) taken prior to acquiring our SIMS data. The indicated analysis numbers correspond to those in Supplementary Data 1. The indicated ages are the radiogenic $^{207}Pb/^{206}Pb$ ages for the corresponding craters.

## Results

**U–Pb systematics.** A single trachyte sample (MAU-8; location given in inset to Fig. 1) was selected for zircon separation based on available sample mass ($\sim 1$ kg) and Zr concentration (1,165 $\mu$g g$^{-1}$); extreme care was taken during sample processing to avoid any risk of contamination. Thirteen zircon grains were recovered, from which we report 68 individual point analyses acquired using the Cameca 1280-HR SIMS (secondary ion mass spectrometer) instrument at GFZ Potsdam; details of the sample processing and analytical methods are provided in the Methods section. Three zircon grains show uniquely mid- to late-Archaean U–Pb systematics, with no evidence for Phanerozoic components. Cathodoluminescence (CL) and backscattered electron (BSE) images (Fig. 2) show that these three crystals contain internal structures such as metamict cores, partially resorbed idiomorphic banding and numerous mineral inclusions. We conducted 20 individual U–Th–Pb spot analyses on these grains, where the concordant or near-concordant $^{207}$Pb/$^{206}$Pb ages range from $3,030 \pm 5$ Ma to $2,552 \pm 11$ Ma (Fig. 3 and Supplementary Data 1). We interpret these data as indicating the crystallization ages of a complex Archaean xenocrystic component; no over-growths of young zircon are evident in any of the three Archaean grains, despite great efforts to identify such rims. The mineral inclusions in these ancient zircons include quartz, K-feldspar and monazite, as determined by EDS measurements; this assemblage would be consistent with crystallization from granitic or syenitic magmas. As the three grains are distinct in terms of their crystallization ages as well as their Th/U systematics, we conclude that trachytic magma traversed through and incorporated silicic continental crustal material that preserves a record of several hundred Myr of Archaean evolution. This is consistent with Sr–Nd isotopic systematics of the Mauritian trachytes[7], as discussed above.

Ten of the 13 grains differ from the older zircons in that they are featureless, with no internal structures visible in CL or BSE images (Fig. 4); these were determined to have late Miocene U–Pb systematics with no traces of inherited components, yielding an unweighted mean $^{206}$Pb/$^{238}$U age of $5.7 \pm 0.2$ Ma (1 s.d., $N = 48$, Fig. 3). We interpret these results as representing the crystal-lization age of the MAU-8 trachyte magma, which is consistent with their association with the Older Series of Mauritian basaltic volcanism (9.0–4.7 Ma, as constrained mainly by $^{40}$Ar/$^{39}$Ar dating[6]).

**Th/U ratios.** U and Th concentrations of all measured individual spots in the zircon grains are given in Supplementary Data 1. For the three Archaean grains, Th/U ratios show distinct values, with Grains 3 and 5 yielding Th/U between 0.05 and 0.47, which is typical of variably metamorphosed and/or recrystallized igneous rocks[9,10]. In contrast, Grain 8 is very depleted in Th, with consistently low Th/U values around 0.01, more typical of high-grade metamorphic rocks[11,12]; we note that among the Archaean zircons, Grain 8 contains the youngest $^{207}$Pb/$^{206}$Pb spot age ($2,552 \pm 11$ Ma, excluding analyses that partly overlapped epoxy or mineral inclusions). In Grains 3 and 8, there is a tendency for spots with lower Th/U to occur on grain margins. These features suggest that the Archaean zircons were derived from source rocks that experienced a complex history of magmatic and metamorphic events. The young (Late Miocene) zircon grains all show much higher Th/U ratios of 0.7–2.6 (average Th/U = $1.4 \pm 0.6$, $N = 48$) typical of magmatic zircons in alkaline rocks[10,13].

**Oxygen isotopes.** In order to further characterize the sources of the zircons, we undertook spot analyses of oxygen isotopes by

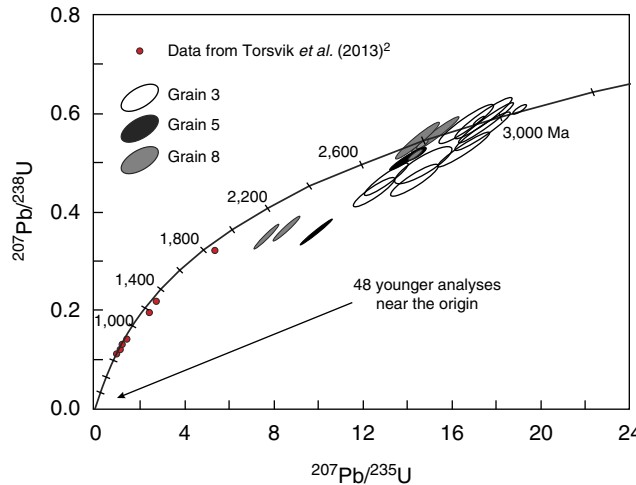

**Figure 3 | Concordia plot.** Includes all 20 data points from the three Archean zircons found in the MAU-8 trachyte sample. The ellipses indicate the 1 s.d. analytical uncertainties for each of the SIMS determinations. Also shown as red symbols are the TIMS wet chemical results reported by Torsvik et al.[2] for eight Proterozoic zircons recovered from Mauritian basaltic beach sands. Not shown are the 48 SIMS determinations on the 10 Miocene zircons, here indicated with the arrow. Tick marks on the Concordia curve are in Ma. TIMS, Thermal Ionization Mass Spectrometry.

SIMS for some of the grains (Supplementary Data 2). For the Archaean zircons, the results yield $\delta^{18}$O$_{SMOW}$ values between 5.5‰ (Grain 3) and 9.7–9.9‰ (Grain 8). The lower value is comparable to the average $\delta^{18}$O$_{SMOW}$ value of $5.5 \pm 0.4$‰ for Archaean zircons in tonalite–trondhjemite–granodiorite rocks in the Superior Province of Canada[14]. The higher values are outside the range of typical $\delta^{18}$O$_{SMOW}$ for magmatic zircons[15], although grains with $\delta^{18}$O$_{SMOW} > 8.5$‰ have been reported in Precambrian granitoids from Fennoscandia[16]. In contrast, oxygen isotope data for 29 spot analyses of the Miocene age zircons give a mean $\delta^{18}$O$_{SMOW}$ value of $4.59 \pm 0.20$‰ (1 s.d.) (Supplementary Data 2). These values are lower than the average $\delta^{18}$O$_{SMOW}$ of mantle zircons ($5.3 \pm 0.3$‰; ref. 15), but are within the range of zircons recovered from some kimberlites ($\delta^{18}$O$_{SMOW} = 3.4$–4.7‰; ref. 17). This would be consistent with an origin for Mauritian trachytes as low-degree partial melts of fertile metasomatized mantle[7].

**Comparison of U–Pb ages with adjacent continents.** Can the spectrum of U–Pb ages for old Mauritian zircons be correlated with exposed Precambrian terranes in nearby continental entities? We considered major continental masses like India, as well as large (e.g., Madagascar, $587 \times 10^3$ km$^2$) and smaller (e.g., Seychelles, 459 km$^2$) continental fragments as potential correla-tives of the sub-Mauritius continental crust. Granitoid rocks of the Seychelles[18] range from $\sim 700$–800 Ma, with the vast majority of ages $\sim 750$ Ma (ref. 19); no Archaean components have been identified there. The Dharwar Craton of southern India (Fig. 1) consists of a nucleus of Palaeoarchaean to Neoarchaean (3.4–2.5 Ga) migmatitic orthogneisses flanked by juvenile Neoarchaean (2.7–2.5 Ga), dominantly granitoid gneisses[20]. Palaeo- and Mesoproterozoic rocks are present south of the Dharwar Craton in India[21]. Neoproterozoic igneous or metaigneous rocks are rare to absent in this region, although the $\sim 750$ Ma Malani Igneous Suite of Rajasthan, some 1,000 km to the NNW, has been correlated with the granitoids of the Seychelles[22,23]. The best match to the age spectrum of Precambrian zircons recovered from Mauritius occurs in east-

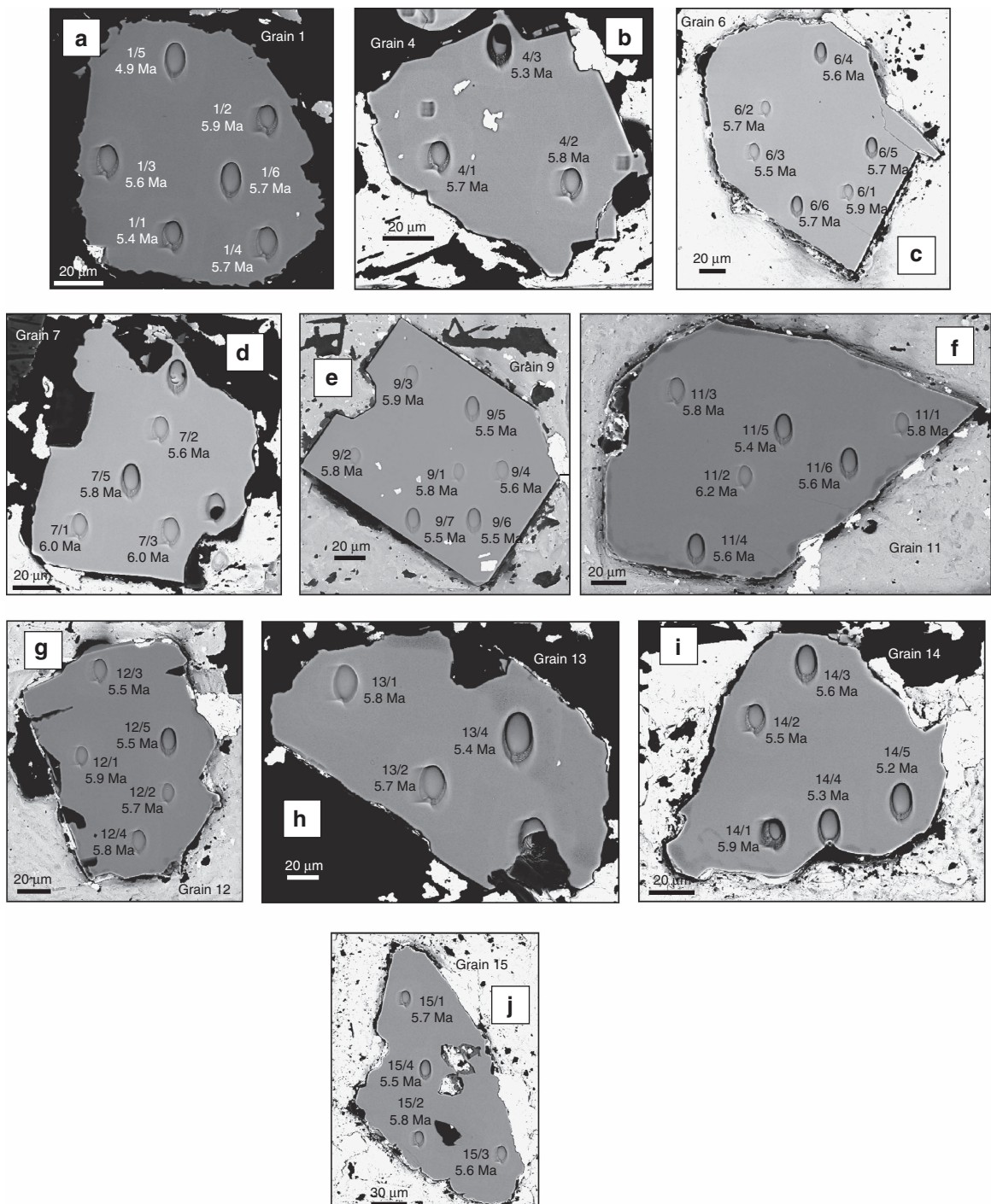

**Figure 4 | Backscattered electron images of the 10 Miocene age zircons recovered from the MAU-8 sample.** Analysis numbers in BSE images (**a**–**j**) correspond to those given in Supplementary Data 1. The indicated ages are the radiogenic $^{206}$Pb/$^{238}$U ages for the corresponding crater.

central Madagascar, where Palaeoarchaean to Mesoarchaean (3.4–3.1 Ga, Antongil–Masora domain) gneisses have been correlated with lithostratigraphic units in the Dharwar Craton, and juvenile Neoarchaean rocks of the Antananarivo domain can be correlated with similar belts in the Eastern Dharwar[24,25]. In addition, gabbroic, syenitic and granitic rocks of Neoproterozoic age (0.85–0.70 Ga) are well represented in Madagascar as the Imorona–Itsindro Suite[24,25], which has been suggested to represent the products of an Andean-type continental arc[26,27]. The entire Precambrian zircon age spectrum found in young Mauritian volcanic rocks could have been derived from a

$\sim 2,000$ km$^2$ area in east-central Madagascar, as illustrated in Fig. 5 (inset map).

## Discussion

Microcontinental fragments are a natural consequence of plate–plume interaction[1,2,28]; an obvious example is the Seychelles, where evidence from surface exposures[18] and seismic studies[29,30] support the existence of a partly emergent $4–4.5 \times 10^4$ km$^2$ fragment of continental crust with a thickness of $\sim 33$ km. Less clear examples include the Laxmi Ridge (Fig. 6), which has been interpreted as a sliver of thinned continental crust[31,32], or

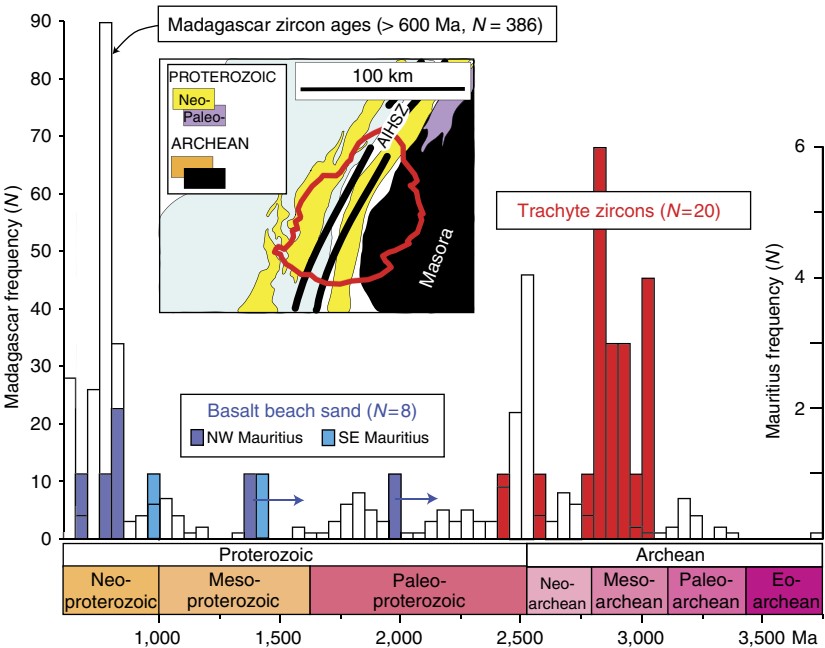

**Figure 5 | Precambrian timescale and histogram of U–Pb ages from Madagascar and Mauritius.** The diagram includes 368 U–Pb ages older than 600 Ma from Madagascar[25], Mauritius trachyte U–Pb ($^{207}Pb/^{206}Pb$) ages of the present study (red bars) and ages from an earlier study of beach sand in Mauritius[2] (blue bars). Arrows indicate that U–Pb ages are minimum ages (Pb-loss). Left-hand scale frequency for Madagascar ages and right-hand frequency ($N$ = number of ages) for Mauritius ages. Inset shows geological map of Madagascar in the Masora region (see Fig. 1, main text) depicting the current size of Mauritius (thick red line); this illustrates that if underlain by continental crust of Madagascar affinity, Mauritius could well sample a spectrum of Neoproterozoic and Archean zircon xenocrysts. AIHSZ, Angavo-Ifanadiana High-Strain Zone in central-east Madagascar.

composed entirely of underplated oceanic crust[33]. In the Indian Ocean, contiguous continental crust of thickness > 25–30 km beneath the Seychelles and northern Mascarenes, and extending south-westwards towards Mauritius (Fig. 6) has been predicted[2], but admittedly crustal thickness determinations from gravity inversion alone cannot distinguish thinned continental crust from anomalously thick oceanic crust. For Mauritius, receiver function data were used[34] to infer a Moho depth of ~ 21 km; those authors favoured entirely oceanic crust thickened by magmatic underplating, although they could not rule out the presence of 'embedded' continental crustal relics. Clearly, geophysical data, in isolation, yield equivocal interpretations of oceanic crustal structures and compositions, but if used in conjunction with geochemical and/or isotopic data (e.g., ref. 7), or with recovered mineral grains that demand the presence of deep continental rocks (this study), the results are more secure.

Our findings confirm the existence of continental crust beneath Mauritius and document, for the first time, the presence of Archaean zircons as xenocrysts in young volcanic rocks from ocean basin settings. Archaean zircons (~ 3,000 Ma) were reported in young lavas and beach sands from the Galapagos, but so far only mentioned in a conference abstract[35]. Younger zircons (~ 330 and ~ 1,600 Ma) have previously been described from Mid-Atlantic Ridge gabbros[36], and Proterozoic (~ 1,800 Ma)[37] and Mesozoic zircons (126–242 Ma)[38] have been found in young Icelandic basalts. Proterozoic zircons (2,430, 1,279, 807 Ma) were recovered in an early Paleogene (64.9 Ma) trachyte from Silhouette Island, Seychelles[39], a result that is relevant to our work because it further improves our understanding of the nature and distribution of Precambrian continental fragments in the Indian Ocean. We propose that Mauritius and other potential Mauritian continental fragments, collectively named Mauritia[2] (Fig. 1; see present location in Fig. 6), are dominantly underlain by Archaean continental crust, and formed part of the ancient nucleus of Madagascar and India.

Conversely, the Seychelles (and probably also the Laxmi Ridge) is likely underlain by Palaeoproterozoic rocks[39], as in northern Madagascar, but was later affected by extensive Neoproterozoic arc magmatism[18].

Mauritia acted as a buffer zone between the western Indian subcontinent and eastern Madagascar, and was fragmented by numerous tectonic and volcanic events that occurred in that region since the early Cretaceous. Due to the early opening between India and East Antarctica in the Enderby Basin, a sinistral shear zone must have acted between India and Madagascar, with India displaced northward relative to Madagascar. That caused convergence and perhaps subduction between northwestern India and Somalia/Arabia[40]. Between 100 and 90 Ma, those displacements were reversed. Soon afterwards, a late Cretaceous Large Igneous Province event (linked to the Marion plume) occurred from about 92 to 84 Ma. It is noteworthy that hardly any volcanism related to the early break-up is found in western margin of India (except the 91.2 Ma St Mary Islands[41]), while most of Madagascar was covered with flood basalts. This suggests that Madagascar and western India were separated at that time by at least ~ 200 km (Fig. 1), a region occupied by Mauritia. The bulk of this continental separation probably existed back in Precambrian times, although Late Jurassic (Tithonian) volcanism in the Seychelles, North Madagascar and Southern India may have been related to a failed rift system[39] that contributed to continental extension between Madagascar and India. Cretaceous pre-breakup strike-slip faulting occurred immediately to the east of Madagascar (blue stippled line in Fig. 1), followed by the opening of the Mascarene Basin, when India, together with Mauritia and the Seychelles/Laxmi Ridge, broke away from Madagascar. Mauritia was subsequently fragmented into a ribbon-like configuration by a series of mid-ocean ridge jumps[2], triggered by the proximity of the Marion, and thereafter the Réunion plume. Subsequent volcanism blanketed most of the

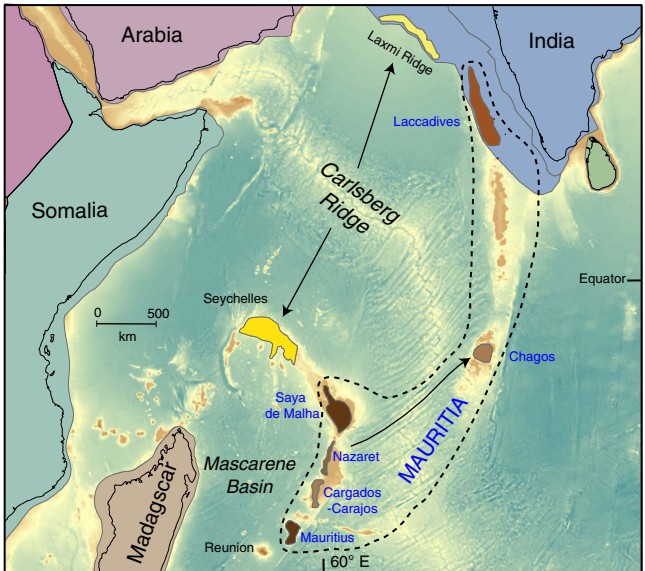

**Figure 6 | Location of possible continental fragments in the Indian Ocean.** These include Mauritia (brownish shading), Laxmi Ridge and the Seychelles (yellowish shading). During the opening of the Mascarene Basin at about 84 Ma, India, together with most of Mauritia and the Seychelles/Laxmi Ridge, broke away from Madagascar. Mauritia was subsequently fragmented into a ribbon-like configuration because of a series of mid-ocean ridge jumps[2], which were partly related to the Marion plume and later the Reunion plume (after 66 Ma). The current configuration with Mauritius/Cargados-Carajos Banks/Nazareth/Saya de Malha forming the Southern Mascarene Plateau (part of the African/Somali Plate), and with the Laccadives and Chagos being part of the Indian Plate, arose at 41 Ma (black arrow shows where Chagos originated at 41 Ma). North of Mauritia, seafloor spreading was initiated between the Laxmi Ridge and the Seychelles at around 62–63 Ma, and the Seychelles became part of the African/Somalian plate after 61 Ma when seafloor spreading in the Mascarene Basin ceased[2]. Background bathymetry is ETOPO1[47] and continental plate polygons with continental-ocean boundaries are from Torsvik & Cocks[48].

Mauritian continental fragments, and in the case of Mauritius, the ancient materials were sampled by plume-related volcanic rocks.

## Methods

**SIMS U–Pb methods.** A key objective of this investigation was to establish whether any older components were present within the late Miocene Mauritian trachytic rocks. The selected sample (MAU-8, Fig. 1) was collected *in situ* from outcrop located at 20.32840° S; 57.71312° E (ref. 7). Approximately 1 kg of material was sent to GFZ Potsdam for sample processing and zircon recovery. Prior to processing, both the sample and the entire crushing facility were carefully cleaned and inspected in order to preclude any risk of cross-contamination from previously processed samples. The MAU-8 sample was friable, so the initial crushing was accomplished by hand using a metal rolling pin on a steel surface. No crushing or grinding apparatus was used. Repeated crushing and sieving resulted in nearly all of the material being reduced to the $<500\,\mu m$ grain size fraction. The resulting material was then panned in tap water, both to remove fine-grained dust and to concentrate the heavy mineral fraction. The dried material was passed through a Frantz magnetic separator, followed by concentration using bromoform, and then methyl iodide heavy liquids. The resulting material was washed, and ultimately 13 zircons could be recovered from the starting material. We exclude any possibility of having contaminated sample MAU-8 with foreign zircons during laboratory procedures: during the preceding decade, no samples of Archaean age had ever been processed in GFZ Potsdam, and the vast majority of material that had been crushed and concentrated was of Phanerozoic age.

The 13 grains were cast in Epofix 2-component cold-set epoxy, along with the 91500 and Temora2 zircon reference materials[42,43]. The sample was then polished to a flatness of $<5\,\mu m$, as confirmed using white light profilometry. The 1 inch diameter sample block was then twice cleaned for 5 min using a high-purity ethanol ultrasonic bath, prior to carbon coating. We used the Potsdam Zeiss Ultra 55 plus field emission scanning electron microscope to image each of the 13 grains

in both backscattered electron and monochromatic cathode luminescence modes. The sample was again cleaned in high-purity ethanol prior to being argon sputter coated with a 35-nm-thick, high-purity gold film. The sample was then placed in a low magnetic susceptibility SIMS sample holder, being held in place with tension springs. The sample was then placed in the airlock of the Cameca 1280-HR SIMS instrument.

Our initial analyses employed a 1–3 nA $^{16}O_2^-$ primary beam, employing Köhler illumination, and focused to $\sim10\,\mu m$ diameter at the surface of the sample. Oxygen flooding, at a pressure of $\sim2\times10^{-3}$ Pa, was used in order to improve Pb sensitivity. The mass spectrometer was operated in mono-collection mode, using a mass resolution of $M/\Delta M\sim5,000$, using an ETP 133H electron multiplier for ion detection, to which a synthetic 46.2 ns dead time was applied to the preamplifier circuit. A single analysis took 32 min, and the data were mostly collected in fully automated mode, using the Cameca point-logger software package. The U–Pb fractionation factor was defined using the Pb/UO versus UO/UO$_2$ relationship, employing a power law fit as defined using the 91500 reference material, which has a $^{206}Pb/^{238}U$ age of 1062.4 Ma (ref. 42). A total of $N=29$ determinations were conducted on the 91500 calibrant during the initial U–Pb session (3 days spread over a week), yielding an unweighted mean $^{206}Pb/^{238}U$ age of 1062.3 ± 6.6 Ma (1 s.d.). We confirmed the accuracy of the U–Pb fractionation correction by measuring $N=21$ determinations on the Temora2 zircon Quality Control Material. Here we determined a mean, unweighted $^{206}Pb/^{238}U$ age of 421.1 ± 2.8 Ma (1 s.d.), which is in reasonable agreement with the published age of 416.50 Ma (ref. 43), confirming that no significant systematic bias is present in our U/Pb determinations. Common Pb corrections, where needed, were based on the observed $^{204}Pb/^{206}Pb$ ratio (older grains) or $^{207}Pb/^{206}Pb$ ratio (younger grains), in conjunction with a recent common Pb composition.

Our initial survey of U–Pb ages identified three grains with Archaean ages. We decided to survey the two larger of these grains (Grains 3 and 8, Fig. 2) in detail, using a small beam diameter. A 100 pA $^{16}O_2^-$ primary beam employing a Gaussian beam distribution was focused to a $\sim2\,\mu m$ diameter at the sample surface; this lower primary beam current resulted in a much lower ion collection rate. In order to produce a flat-bottomed sputtering cater, a $5\times5\,\mu m$ raster was applied to the primary beam, and this was compensated for using the dynamic transfer capability of the 1280-HR's secondary ion optics. All other aspects of the second data acquisition series were the same as those for earlier 2 nA analyses. Measurements of the Temora2 Quality Control Material gave an unweighted $^{206}Pb/^{238}U$ age of 410 ± 5.3 Ma (1 s.d., $N=6$); despite the low primary ion current used, this is in reasonable agreement with the assigned 416.50 Ma age[43] of the Temora2 zircon suite. After completion of this second analytical series, we again imaged the sample in BSE mode using the Potsdam SEM, thereby documenting the exact locations of each analysis point (Figs 2 and 4). This imaging required that an additional 20 nm of gold be sputter coated onto the sample mount. Finally, SIMS crater dimensions were determined using a Zygo Lot 7100 white light profilometer. These reveal crater depths of 2.7 and 1.3 μm for the 2 nA and 100 pA analyses, respectively. From the profilometry results, we estimate test portion masses of $\sim3.2$ ng and $\sim150$ pg for the two machine settings, respectively.

Results of all SIMS spot analyses of zircons in trachyte sample MAU-8 are given in Supplementary Data 1, and are plotted on a conventional Concordia diagram in Fig. 3, along with the thermal ionization mass spectrometry analyses of Proterozoic zircons recovered from basaltic beach sands[2]. All results for Precambrian zircons from Mauritius are plotted on a histogram in Fig. 5, along with a compilation of Precambrian zircons from Madagascar ($N=386$)[25].

**SIMS oxygen isotope methods.** With the intent of further characterizing the nature of our suite of Mauritian zircons, we undertook additional oxygen isotope determinations by SIMS. Our analytical technique closely followed that described for the Potsdam SIMS laboratory[44], using the 91500 zircon reference material for calibration, which has an assigned $\delta^{18}O_{SMOW}$ value of 9.86 ± 0.11‰, based on gas source mass spectrometry[45]. Due to the fact that the zircons were both small in size and thin, due to earlier sample polishing, it was decided that a repolishing of the mount was imprudent. Furthermore, due to the numerous U–Th–Pb determinations, which had been previously conducted and which would have implanted $^{16}O$ from the SIMS primary beam, there were only three locations on the Archean grains that could be analyzed. These results, along with the results from 29 locations on Miocene zircons, are given in Supplementary Data 2.

**Imaging methods.** Backscattered electron images show the presence of numerous mineral inclusions in 2 of the 3 Archaean grains (Fig. 2), whereas all 10 of the Miocene grains were inclusion-free (Fig. 4). In order to identify the nature of these inclusions, we removed the gold coating by gently rubbing with an ethanol-saturated tissue. The mount was then cleaned in high-purity ethanol for 5 min using an ultrasonic bath. The sample was sputter coated with carbon, followed by EDX analyses using the GFZ Potsdam Zeiss SEM instrument. From the major element abundances we identified quartz, K-feldspar and monazite.

**Data availability.** The authors declare that all relevant data are available within the article and its Supplementary Data files. Other pertinent data are available from the authors upon request.

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

## Acknowledgements

Zircon extraction in Potsdam was overseen by J. Glodny. F. Couffignal was responsible for the SIMS data acquisition parts of this project. A. Rocholl and Z. Jin supported the data reduction and presentation of the U–Pb results. I. Schäpan provided both the SEM images and the EDS chemical analyses. The Research Council of Norway, through its Centres of Excellence funding scheme, project number 223272 (CEED), and the European Research Council, under the European Union's Seventh Framework Programme (FP7/2007–2013/ERC Advanced Grant Agreement Number 267631, Beyond Plate Tectonics), are acknowledged for financial support. L.D.A. acknowledges support from the National Research Council (South Africa) and the University of the Witwatersrand. We thank Chris Harris for advice on oxygen isotope interpretations, and Carmen Gaina, Alan Collins, Greg Shellnutt and Jean-Louis Paquette for constructive and helpful comments.

## Author contributions

L.D.A. collected and characterized the samples petrologically and geochemically, and initially recovered the first zircons in the trachytic rocks. M.W. oversaw the SIMS and SEM laboratory experiments, and played the leading role in data reduction and the interpretation of the SIMS isotopic data. T.H.T. provided the plate reconstructions. All authors contributed to the manuscript.

## Additional information

**Competing financial interests:** The authors declare no competing financial interests.

**Publisher's note**: 

