## [Peer Review File · Nature Communications]

Reviewer #1 (Remarks to the Author):

Thanks for the opportunity to look at this interesting report of ancient zircons from the trachytes in Mauritius. The major claim of this paper is that these are the first Archaean zircons recovered from young rocks from the oceanic lithosphere. Although Proterozoic zircons have been previously reported from beach sands in Mauritius and similar examples have been recently reported at the IGC from the Galapagos (Rojas-agramonte et al. IGC Cape Town abstract), to my knowledge there are no in-situ Archaean zircons reported previously from this environment. This is significant as it refutes the possibility of pumice rafting that is present when only beach sand zircons have been discovered. The authors go further here and propose that the ages so far recovered from both the trachytes and from the Mauritian beach sands suggest a strong link with the sub-surface of Mauritia (as they call this palaeocontinent) and SE Madagascar. I find this certainly plausible and although the authors sometimes use old references, where more recent comprehensive ones would be better, to back this up, I do find this extremely interesting and of general significance.

The isotopic data from the three zircons could be more convincing, and presented more completely (for example quoting MSWDs for means to allow an assessment of the accuracy), but the zircons certainly are Archaean. I would like to see CL images presented for the three key zircons. Especially because the most convincing grain (grain 3) has a range of concordant data that is too large to be simply due to analytical error. So what has caused this single grain to yield concordant, or at least pseudo-concordant (due to the imprecision of the SIMS technique), results? Are there discrete CL domains with ca. 3.1 Ga zircon and younger domains (or even a rim?) with ca. 2.8 Ga zircon? This could better constrain a possible origin as an interpretation of this could be that the zircon formed in a magma at ca. 3.1 Ga and was metamorphosed at ca. 2.8 Ga.

I very much like Fig. 1, and have suggested some improvements to ensure its accuracy.

Overall, I think that this is an excellent, interesting and potentially quite important paper for understanding both the Proterozoic evolution of this area and the breakup of Gondwana and formation of the Indian Ocean.

I have some other comments and suggestions on the attached annotated manuscript

Sincerely
Alan Collins

Reviewer #2 (Remarks to the Author):

Review of First Archaean zircons found in oceanic crustal rocks

The paper presents new in situ SIMS U/Pb ages of zircons extracted from a Miocene trachyte that intruded the Older basaltic volcanic series of eastern Mauritius. Three zircons yielded Mesoarchean ages (3.0 Ga to 2.8 Ga) whereas the majority yielded ages that are interpreted to represent the age of the host rock. The authors suggest that the Archean zircons along with gravity anomaly data and the identification of Proterozoic zircons from beach sands are evidence for the existence of continental crust beneath Mauritius. Specifically, the crust is thought to be derived from central-east Madagascar. The results imply that the continental crust from eastern Madagascar and western India was fragmented and dispersed across the western Indian Ocean during the Cretaceous to Cenozoic break-up of remnant East Gondwana.

The impact of the work is identifying the extent of crustal fragmentation during rifting and its subsequent impact on hotspot magmatism. The paper is well written and the results are quite interesting. The authors convincingly make a case for the existence of continental crust beneath Mauritius but the precise extent is not examined. The paper can be accepted in its current form

although I think it would be beneficial to expand on processes of fragmentation and rifting vis-à-vis the role mantle-plumes may play. For example there are a number of submerged plateaux in the Indian Ocean that are closely associated with rifted margins of East Gondwana. It is likely that some of them may also be underlain by continental crust. However, in comparison, the Atlantic Ocean appears to have fewer plateaux, particularly in the middle and south. Why does the Indian Ocean have more plateaux? Is it due to the presence of more cratons or suture zones? Is it related to mantle-plume-induced fragmentation? Were the suture/rift zones of East Gondwana more difficult to exploit? Did the Indian Ocean experience more ridge jumps than the Atlantic Ocean? Including ruminations on some of these questions would be appreciated.

Also, we identified a very small population of Tithonian (~147 Ma) zircons from Seychelles that are contemporaneous with rocks from northern Madagascar and southwest India. We suggested that there may have been magmatism related to an aulacogen that extended from Seychelles/India along what will become the Late Cretaceous rifted margin of India-Madagascar. I freely admit that the evidence for a Tithonian aulacogen is limited at this point but it could be a contributing factor in the distance (~200 km) separating Madagascar and western India during the Late Cretaceous. As it is, the paper is not particularly clear why shearing (trans-tension?) would cause a ~200 km separation between Madagascar and India

Greg Shellnutt
National Taiwan Normal University

Reviewer #3 (Remarks to the Author):

Review of the manuscript NCOMMS-16-22428-T entitled "First Archean zircons found in oceanic crustal rocks" submitted by L.D. Ashwal and co-workers at Nature Communications.

The manuscript: "First Archean zircons found in oceanic crustal rocks" is a very important and interesting paper which provides significant new data and concept on the understanding of the evolution of the oceanic lithosphere. I strongly appreciated these results and the paper and I suggest to publish it with minor revisions which are listed after the following questionnaire.

What are the major claims of the paper?

There are two major claims. The most important is the demonstration of the occurrence of remnants of an Archean continental crust beneath Mauritius oceanic island demonstrated by the occurrence of Archean xenocrystic zircons sampled by Miocene volcanic rocks. This confirms that at least the Mauritius island and maybe more oceanic islands were formed and anchored on a fragment of Precambrian continental crust. The second claim is the occurrence of Archean zircons in oceanic crustal rocks as described by the title. Even though important, this finding is not as important as the first one, because many occurrences of inherited zircons were already found in past ocean crust-derived ophiolites and eclogites demonstrating their emplacement and crystallization in the vicinity of continental rocks.

Are the claims novel?

Yes it is. Precambrian zircons were already found in beach sands from Mauritius island by some of the authors allowing them to propose the hypothesis of a Precambrian micro continent in the Indian Ocean. Here, these zircons were incorporated into the trachytes possibly as xenoliths or xenocrysts during the ascent of the lava or into the magma chamber during its formation. I think this is the first time that such a finding happens in an oceanic island context.

Will the paper be of interest to others in the field?

Of course, this is very important for the global geodynamics of the Indian Ocean and further to the formation of oceanic islands or plateaux.

Will the paper influence thinking in the field?

Of course, it is not usual to consider continental crust within "pure" oceanic domains. It also raises questions about the systematic occurrence of some enriched oceanic mantle reservoirs which could be debated in the light of the preservation of continental crust into oceanic domains.

Are the claims convincing?

Undoubtedly yes. The occurrence of these Archean zircons is at the same time very simple and very strong.

Are there other experiments that would strengthen the paper?

Actually no because new Precambrian zircons will not provide many more constraints. In the future, I am convinced that this finding will be confirmed in other places.

Are the claims appropriately discussed in the context of previous literature?

Yes, particularly the geodynamic model.

The manuscript is acceptable with minor revisions and should not be re-submitted.

Is the manuscript clearly written?

Yes

Could the manuscript be shortened?

No, I rather suggest a few more discussions.

Have the authors done themselves justice without overselling their claims?

Yes, I think so. This is a very important finding with really strong consequences.

Have they been fair in their treatment of previous literature?

Yes

Have they provided sufficient methodological detail?

Yes

Is the statistical analysis of the data sound?

Yes, I have minor remarks on that point.

Should the authors be asked to provide further data or methodological information?

Not necessary.

Following I listed more precise remarks and requirements.

- First, I do not find your title really informative and strongly connected to your finding. Of course you dated Archean zircons in oceanic crustal rocks but this was already published on many depleted-mantle rocks sampled in old ophiolite and eclogite domains from the Variscan belt or the Alps for example. Consequently, you should mention that your oceanic rocks are recent: Neogene or Miocene. Additionally, the term "oceanic rocks" is rather understood as depleted-mantle derived MORB-like rocks than trachytes from oceanic island. It is not false but the reader means what you call oceanic crust after discovering the whole paper. So, I suggest to focus the title on the main finding: the occurrence of Archean continental crust and zircons beneath the Mauritius oceanic island.

- p.3 line 57, you report Zr content of 1165 ppm for sample MAU8. This very high whole rock Zr content associated with very high (>2000 ppm) Th content and >1 Th/U ratios of the zircons from the trachyte is frequent in the zircons derived from metasomatically-enriched subcontinental lithospheric mantle. This point should be discussed.

- p.3 line 62: you report cathodo-luminescence images but only "full white" BSE images are

available on figure 2 and extended data Figure 1. Considering the importance of these zircon crystals, I think that figures associating cathodo-luminescence and BSE images are absolutely required.

- p.3 line 66: you refer to Extended data Table 1 where Th and U contents are reported but not Th/U ratios. I calculated these Th/U ratios which are within a typically 0.2-0.5 magmatic range for grains 3 and 5 but typically metamorphic value of 0.01 for grain 8. That means that your three Archean grains report a complex history with composite magmatic and metamorphic ages and corresponding sources which should be pointed out. Also for that reason cathodo-luminescence images may illustrate differences in the inner structures of both zircon types.

-p.3 line 69: you show a monazite inclusion into the metamorphic zircon grain. Such small inclusion could be dated using a 2-3 μm spot with LA-ICPMS and may provide interesting constraint.

-p.3 line 72: you claim that the trachyte magma traversed and incorporated silicic continental crust material, which is probable. In the manuscript at Journal of Petrology you also consider that small amount of old continental crust could be incorporated to the magma source, which is very possible. An inherited zircon crystal can remain in equilibrium with a trachytic melt without dissolving. Consequently, both origins for those zircon xenocrysts are possible and I suggest to mention it.

-p.3 line 78: I absolutely do not understand that statistic treatment because you are dating an unknown sample and not evaluating a new mineral reference material. Consequently, I agree that the unweighted mean and the standard deviation of the Gaussian curve corresponding to the $^{206}\text{Pb}/^{238}\text{U}$ ages is 5.7 ± 0.2 Ma, but this is not the calculation generally operated by the geochronology community. Two possibilities are available; the first one is the weighted mean of the $^{206}\text{Pb}/^{238}\text{U}$ ages quoted at the 95% confidence level for univariate normal distribution yielding 5.66 ± 0.05 Ma considering 47 analyses (3 outliers) and a MSWD of 2.6 (see diagram below). The best possibility would be to report the dataset in a Tera & Wasserburg diagram (as an extended data figure in the appendix) and the corresponding lower intercept. Considering that the $^{207}\text{Pb}/^{206}\text{Pb}$ ratios are measured and used for common Pb correction, this is easy to do.

-p.3 line 80. Another trachyte sample VEN from another location is reported at Mauritius and yield an age of 6.85 ± 0.01 Ma (see your paper in press at the Journal of Petrology). Both samples with apparently similar source and petrogenesis provide significantly different ages, this should at least be noted in this paper.

-p.4, lines 117-118. This sentence would be a good title for the paper. Maybe you could stress on the point that at the time of emplacement of the trachyte sample, 5.7 Ma ago, the Indian and Madagascar continental blocks were very far from Mauritius. Consequently, a contamination of the trachyte by zircons coming from their sediment products is excluded.

-p.5-6 lines 166, 188 and 188: there are two mistakes on the reference numbers in the methods; references numbered 29 and 30 should be replaced by 32-33.

-p.6 line 179: I know the dead time for an amplifier and an idle time for a magnet but I don't know what is called there a "dreamtime".

-p.6 lines 185 and 187: In references 32 and 33, the 91500 reference material is reported with $^{206}\text{Pb}/^{238}\text{U}$ age weighted mean with 1s uncertainty and Temora 2 reference material with $^{206}\text{Pb}/^{238}\text{U}$ age weighted mean with 2s uncertainties. The measurements of the same reference materials performed in the Potsdam laboratory are detailed in this paragraph as unweighted means at the 1 s level. These statistics are too different from one to the other. In agreement with many members of the geochronology international community (see Horstwood et al., 2016,

Geostandards and Geoanalytical Research, 40, pp. 311-332), please quote all your uncertainties in the text, diagrams and tables at the 2 sigma level.

-p.6 line 189. I agree that no significant bias is present in the U/Pb determinations but the key results of this paper are reported in p.3 as mean $^{207}\text{Pb}/^{206}\text{Pb}$ ages. Consequently, if the inter-element fractionation is well corrected, you could also provide a few values demonstrating that $^{207}\text{Pb}/^{206}\text{Pb}$ mass bias is also well defined.

-p.16 Extended Data Table 1: I am not really happy with that table (see again Horstwood et al., 2016). (1) I am not sure that the primary current is useful to the reader. (2) Since Th and U contents are reported, an additional column for the Th/U ratios would be interesting. (3) $^{206}\text{Pb}/^{238}\text{U}$ and $^{207}\text{Pb}/^{206}\text{Pb}$ ratios and corresponding uncertainties (preferably 2 s) are required. (4) $^{206}\text{Pb}/^{238}\text{U}$ and $^{207}\text{Pb}/^{206}\text{Pb}$ apparent ages and corresponding uncertainties at the 2 s level are required. (5) Considering the range of ages and errors, digits after the point are useless for Archean zircons. (6) Due to the large errors for old zircons and the very long ^{232}Th decay constant for the young ones, $^{206}\text{Pb}/^{232}\text{Th}$ apparent ages are not necessary.

Jean-Louis Paquette

Dear Melissa,

We have revised our manuscript in accordance with all of the reviewers' comments, as well as yours. We thank you for choosing such appropriate reviewers, whose comments were extremely helpful, and which have resulted in a stronger and better paper. Our detailed responses to each of the reviewers' comments are given below. We have changed the title to better convey our scientific message, without using the term "first". We have lengthened the paper to include some new oxygen isotope data for our zircons, and have expanded the discussion, as suggested by the reviewers. We have adjusted the structure of the paper to follow the outline you provided for papers in Nature Communications. We hope that you will find our submission acceptable for publication, and we thank you again for your prompt and professional editorial handling.

Lew Ashwal

Michael Wiedenbeck

Trond Torsvik

Reviewers' comments:

Reviewer #1 (Remarks to the Author):

Thanks for the opportunity to look at this interesting report of ancient zircons from the trachytes in Mauritius. The major claim of this paper is that these are the first Archaean zircons recovered from young rocks from the oceanic lithosphere. Although Proterozoic zircons have been previously reported from beach sands in Mauritius and similar examples have been recently reported at the IGC from the Galapagos (Rojas-agramonte et al. IGC Cape Town abstract), to my knowledge there are no in-situ Archaean zircons reported previously from this environment. This is significant as it refutes the possibility of pumice rafting that is present when only beach sand zircons have been discovered. The authors go further here and propose that the ages so far recovered from both the trachytes and from the Mauritian beach sands suggest a strong link with the sub-surface of Mauritia (as they call this palaeocontinent) and SE Madagascar. I find this certainly plausible and although the authors sometimes use old references, where more recent comprehensive ones would be better, to back this up, I do find this extremely interesting and of general significance.

We have added more up-to-date references to support our proposed connection to SE Madagascar, including the work of this reviewer's group (Archibald et al., 2015).

The isotopic data from the three zircons could be more convincing, and presented more completely (for example quoting MSWDs for means to allow an assessment of the accuracy), but the zircons certainly are Archaean. I would like to see CL images presented for the three key zircons. Especially because the most convincing grain (grain 3) has a range of concordant data that is too large to be simply due to

analytical error. So what has caused this single grain to yield concordant, or at least pseudo-concordant (due to the imprecision of the SIMS technique), results? Are there discrete CL domains with ca. 3.1 Ga zircon and younger domains (or even a rim?) with ca. 2.8 Ga zircon? This could better constrain a possible origin as an interpretation of this could be that the zircon formed in a magma at ca. 3.1 Ga and was metamorphosed at ca. 2.8 Ga.

We have modified our Table of U-Pb isotopic data (see detailed responses below to Paquette's comments). Our ages are concordia intercept ages, not isochrons, and so there is no need to report MSWD values. We have added CL images to accompany the BSE images of the three Archaean zircons. This was a good suggestion and we are happy to comply.

I very much like Fig. 1, and have suggested some improvements to ensure its accuracy.

We have adjusted Figure 1 according to suggested changes in the PDF file. Specifically the "Paleoproterozoic" has been relabeled to "Paleoproterozoic-Mesoproterozoic" and brown parts in India have been made black as the rest of the Paleoproterozoic. Following Archibald et al. (2015) we have also coloured the Ambatolampy Group in pink. Added reference: Archibald, D.B, Collins, A.S., Fodena, J.D., Payne, J.L., Taylor, R et al. Towards unravelling the Mozambique Ocean conundrum using a triumvirate of zircon isotopic proxies on the Ambatolampy Group, central Madagascar. *Tectonophysics* 662, 167–182 (2015).

Overall, I think that this is an excellent, interesting and potentially quite important paper for understanding both the Proterozoic evolution of this area and the breakup of Gondwana and formation of the Indian Ocean.

I have some other comments and suggestions on the attached annotated manuscript

Sincerely
Alan Collins

Reviewer #2 (Remarks to the Author):

Review of First Archaean zircons found in oceanic crustal rocks

The paper presents new in situ SIMS U/Pb ages of zircons extracted from a Miocene trachyte that intruded the Older basaltic volcanic series of eastern Mauritius. Three zircons yielded Mesoproterozoic ages (3.0 Ga to 2.8 Ga) whereas the majority yielded ages that are interpreted to represent the age of the host rock. The authors suggest that the Archaean zircons along with gravity anomaly data and the identification of Proterozoic zircons from beach sands are evidence for the existence of continental crust beneath Mauritius. Specifically, the crust is thought to be derived from central-east Madagascar. The

results imply that the continental crust from eastern Madagascar and western India was fragmented and dispersed across the western Indian Ocean during the Cretaceous to Cenozoic break-up of remnant East Gondwana.

The impact of the work is identifying the extent of crustal fragmentation during rifting and its subsequent impact on hotspot magmatism. The paper is well written and the results are quite interesting. The authors convincingly make a case for the existence of continental crust beneath Mauritius but the precise extent is not examined. The paper can be accepted in its current form although I think it would be beneficial to expand on processes of fragmentation and rifting vis-à-vis the role mantle-plumes may play. For example there are a number of submerged plateaux in the Indian Ocean that are closely associated with rifted margins of East Gondwana. It is likely that some of them may also be underlain by continental crust. However, in comparison, the Atlantic Ocean appears to have fewer plateaux, particularly in the middle and south. Why does the Indian Ocean have more plateaux? Is it due to the presence of more cratons or suture zones? Is it related to mantle-plume-induced fragmentation? Were the suture/rift zones of East Gondwana more difficult to exploit? Did the Indian Ocean experience more ridge jumps than the Atlantic Ocean? Including ruminations on some of these questions would be appreciated.

Also, we identified a very small population of Tithonian (~147 Ma) zircons from Seychelles that are contemporaneous with rocks from northern Madagascar and southwest India. We suggested that there may have been magmatism related to an aulacogen that extended from Seychelles/India along what will become the Late Cretaceous rifted margin of India-Madagascar. I freely admit that the evidence for a Tithonian aulacogen is limited at this point but it could be a contributing factor in the distance (~200 km) separating Madagascar and western India during the Late Cretaceous. As it is, the paper is not particularly clear why shearing (trans-tension?) would cause a ~200 km separation between Madagascar and India

We did seriously consider discussing/implementing some of the comments above in the paper, but this paper is mostly focused on the geochronology and a realistic plate reconstruction at around 90-85 Ma; and not on “Fundamental processes of fragmentation (rifting vs. the role mantle-plumes) and the importance of pre-existing sutures on a global scale”.

We are also aware of the suggestion of a possible Late Jurassic-Early Cretaceous aulacogen between Madagascar and India – but as the reviewer states – the evidence for this is limited and therefore not something we feel to elaborate/discuss in this paper. Finally, we are not claiming that the 200 km separation of Madagascar and India in the Late Cretaceous was caused by shearing/trans-tension but is the original separation that may have existed since Precambrian times (so they were never juxtaposed along their coastlines as in many existing models). To accommodate the reviewers comment in this paragraph we added the following sentence near the end of the manuscript:

“The bulk of this continental separation probably existed back in Precambrian times but Late Jurassic (Tithonian) volcanism in the Seychelles, North Madagascar and Southern India may have been related

to a failed rift system (Shellnutt et al. 2015) that may have contributed to continental extension between Madagascar and India. “

Greg Shellnutt
National Taiwan Normal University

Reviewer #3 (Remarks to the Author):

Review of the manuscript NCOMMS-16-22428-T entitled "First Archean zircons found in oceanic crustal rocks" submitted by L.D. Ashwal and co-workers at Nature Communications.

The manuscript:"First Archean zircons found in oceanic crustal rocks" is a very important and interesting paper which provides significant new data and concept on the understanding of the evolution of the oceanic lithosphere. I strongly appreciated these results and the paper and I suggest to publish it with minor revisions which are listed after the following questionnaire.

What are the major claims of the paper?

There are two major claims. The most important is the demonstration of the occurrence of remnants of an Archean continental crust beneath Mauritius oceanic island demonstrated by the occurrence of Archean xenocrystic zircons sampled by Miocene volcanic rocks. This confirms that at least the Mauritius island and maybe more oceanic islands were formed and anchored on a fragment of Precambrian continental crust. The second claim is the occurrence of Archean zircons in oceanic crustal rocks as described by the title. Even though important, this finding is not as important as the first one, because many occurrences of inherited zircons were already found in past ocean crust-derived ophiolites and eclogites demonstrating their emplacement and crystallization in the vicinity of continental rocks.

Are the claims novel?

Yes it is. Precambrian zircons were already found in beach sands from Mauritius island by some of the authors allowing them to propose the hypothesis of a Precambrian micro continent in the Indian Ocean. Here, these zircons were incorporated into the trachytes possibly as xenoliths or xenocrysts during the ascent of the lava or into the magma chamber during its formation. I think this is the first time that such a finding happens in an oceanic island context.

Will the paper be of interest to others in the field?

Of course, this is very important for the global geodynamics of the Indian Ocean and further to the formation of oceanic islands or plateaux.

Will the paper influence thinking in the field?

Of course, it is not usual to consider continental crust within "pure" oceanic domains. It also raises questions about the systematic occurrence of some enriched oceanic mantle reservoirs which could be debated in the light of the preservation of continental crust into oceanic domains.

Are the claims convincing?

Undoubtedly yes. The occurrence of these Archean zircons is at the same time very simple and very strong.

Are there other experiments that would strengthen the paper?

Actually no because new Precambrian zircons will not provide many more constraints. In the future, I am convinced that this finding will be confirmed in other places.

Are the claims appropriately discussed in the context of previous literature?

Yes, particularly the geodynamic model.

The manuscript is acceptable with minor revisions and should not be re-submitted.

Is the manuscript clearly written?

Yes

Could the manuscript be shortened?

No, I rather suggest a few more discussions.

Have the authors done themselves justice without overselling their claims?

Yes, I think so. This is a very important finding with really strong consequences.

Have they been fair in their treatment of previous literature?

Yes

Have they provided sufficient methodological detail?

Yes

Is the statistical analysis of the data sound?

Yes, I have minor remarks on that point.

Should the authors be asked to provide further data or methodological information?

Not necessary.

Following I listed more precise remarks and requirements.

- First, I do not find your title really informative and strongly connected to your finding. Of course you dated Archean zircons in oceanic crustal rocks but this was already published on many depleted-mantle rocks sampled in old ophiolite and eclogite domains from the Variscan belt or the Alps for example. Consequently, you should mention that your oceanic rocks are recent: Neogene or Miocene.

Additionally, the term "oceanic rocks" is rather understood as depleted-mantle derived MORB-like rocks than trachytes from oceanic island. It is not false but the reader means what you call oceanic crust after discovering the whole paper. So, I suggest to focus the title on the main finding: the occurrence of Archean continental crust and zircons beneath the Mauritius oceanic island.

We have changed our title to: "Archean zircons in Miocene oceanic hotspot rocks establish ancient continental crust beneath Mauritius". This new title mentions the age of the host rocks of the Archean zircons, their identity as hotspot products, and we have removed the term "first", as stipulated by this review and the Editor.

- p.3 line 57, you report Zr content of 1165 ppm for sample MAU8. This very high whole rock Zr content associated with very high (>2000 ppm) Th content and >1 Th/U ratios of the zircons from the trachyte is frequent in the zircons derived from metasomatically-enriched subcontinental lithospheric mantle. This point should be discussed.

The incompatible element enrichment of Mauritian trachytes was discussed in the recently-published paper in Journal of Petrology (Ashwal et al., 2016), and we do not feel it necessary to repeat these issues here. As suggested by this reviewer, we have added a short discussion on Th/U of our zircons (see below).

- p.3 line 62: you report cathodo-luminescence images but only "full white" BSE images are available on figure 2 and extended data Figure 1. Considering the importance of these zircon crystals, I think that figures associating cathodo-luminescence and BSE images are absolutely required.

This is a good suggestion, and we have added CL images to accompany the BSE images shown in Fig. 2.

- p.3 line 66: you refer to Extended data Table 1 where Th and U contents are reported but not Th/U ratios. I calculated these Th/U ratios which are within a typically 0.2-0.5 magmatic range for grains 3 and 5 but typically metamorphic value of 0.01 for grain 8. That means that your three Archean grains report a complex history with composite magmatic and metamorphic ages and corresponding sources which should be pointed out. Also for that reason cathodo-luminescence images may illustrate differences in the inner structures of both zircon types.

This is also a good suggestion, and we have added a column of Th/U ratios to our Supplementary Table 1. We have also added a short paragraph in which we discuss the interesting Th/U ratios of both the Archean and Miocene zircons. The Th/U ratios support our conclusion that the Archean zircons were derived from source rocks that experienced a complex history of magmatic and metamorphic events, very much like the exposed rocks in SE Madagascar. The young zircons have high Th/U ratios typical of magmatic zircons in alkaline igneous rocks.

-p.3 line 69: you show a monazite inclusion into the metamorphic zircon grain. Such small inclusion

could be dated using a 2-3 μm spot with LA-ICPMS and may provide interesting constraint.

We do not at present have access to LA-ICPMS dating facilities, and in any case we feel that carrying out dating of this single grain would be beyond the scope of the present paper.

-p.3 line 72: you claim that the trachyte magma traversed and incorporated silicic continental crust material, which is probable. In the manuscript at Journal of Petrology you also consider that small amount of old continental crust could be incorporated to the magma source, which is very possible. An inherited zircon crystal can remain in equilibrium with a trachytic melt without dissolving. Consequently, both origins for those zircon xenocrysts are possible and I suggest to mention it.

We feel that the preservation of Archaean ages in zircons that might be present in the mantle source regions of the trachytes is extremely unlikely. Therefore we prefer not to mention this as a possibility. Our paper focuses on the more likely interpretation that the zircons were derived from ancient continental material that underlies Mauritius.

-p.3 line 78: I absolutely do not understand that statistic treatment because you are dating an unknown sample and not evaluating a new mineral reference material. Consequently, I agree that the unweighted mean and the standard deviation of the Gaussian curve corresponding to the $^{206}\text{Pb}/^{238}\text{U}$ ages is 5.7 ± 0.2 Ma, but this is not the calculation generally operated by the geochronology community. Two possibilities are available; the first one is the weighted mean of the $^{206}\text{Pb}/^{238}\text{U}$ ages quoted at the 95% confidence level for univariate normal distribution yielding 5.66 ± 0.05 Ma considering 47 analyses (3 outliers) and a MSWD of 2.6 (see diagram below). The best possibility would be to report the dataset in a Tera & Wasserburg diagram (as an extended data figure in the appendix) and the corresponding lower intercept. Considering that the $^{207}\text{Pb}/^{206}\text{Pb}$ ratios are measured and used for common Pb correction, this is easy to do.

Paquette's requests that we use our data set to calculate a precise value for the eruption age of the trachyte based on the 49 determinations on the younger grains. We are reluctant to do this for two reasons.

Firstly, we cannot confirm that the population, in fact, represents a Gaussian distribution as would be required in order to justify such a calculation. Secondly, with $N = 49$ the uncertainty on the mean value would be, even when mathematically correct, misleadingly precise. It would not adequately consider the presence of systematic biases (for example, an overcorrection of the dark noise on our ion counter) that, albeit small, could have impact on our results on such young ages at a level greater than the analytical uncertainty suggested by a straightforward Gaussian-based uncertainty estimation. As the focus of our work is the older materials, we think our original presentation of the unweighted mean $^{206}\text{Pb}/^{238}\text{U}$ age, along with the observed analytical repeatability is most suitable. This confirms the Miocene age and it also documents the stability of our analytical run without

providing an age value with extremely small uncertainty assignment, which we do not see as justified for comparison by future researchers.

Regarding Paquette's recommendation that we provide a Terra-Wasserburg presentation of the Miocene data, this makes little sense in light of our data set, as stated in the notes to 'Supplementary Table 1', being based on a 207Pb correction. To produce a Terra-Wasserburg presentation we would need to make a common Pb correction based on either 204Pb or 208Pb contents, both of which we evaluated and found to be less robust in terms of likely bias in the common Pb corrections.

-p.3 line 80. Another trachyte sample VEN from another location is reported at Mauritius and yield an age of 6.85 ± 0.01 Ma (see your paper in press at the Journal of Petrology). Both samples with apparently similar source and petrogenesis provide significantly different ages, this should at least be noted in this paper.

We have added a statement that mentions our earlier age determination of 6.85 ± 0.01 Ma, as reported in Ashwal et al. (2016).

-p.4, lines 117-118. This sentence would be a good title for the paper. Maybe you could stress on the point that at the time of emplacement of the trachyte sample, 5.7 Ma ago, the Indian and Madagascar continental blocks were very far from Mauritius. Consequently, a contamination of the trachyte by zircons coming from their sediment products is excluded.

We have changed our title in accordance with this reviewer's suggestion, as mentioned above.

-p.5-6 lines 166, 188 and 188: there are two mistakes on the reference numbers in the methods; references numbered 29 and 30 should be replaced by 32-33.

We have fixed the mistakes in the numbering of these and other references.

-p.6 line 179: I know the dead time for an amplifier and an idle time for a magnet but I don't know what is called there a "dreamtime".

"Dreamtime" has been changed to "dead time". We apologize for this error.

-p.6 lines 185 and 187: In references 32 and 33, the 91500 reference material is reported with 206Pb/238U age weighted mean with 1s uncertainty and Temora 2 reference material with 206Pb/238U age weighted mean with 2s uncertainties. The measurements of the same reference materials performed in the Potsdam laboratory are detailed in this paragraph as unweighted means at the 1 s level. These statistics are too different from one to the other. In agreement with many members of the

geochronology international community (see Horstwood et al., 2016, *Geostandards and Geoanalytical Research*, 40, pp. 311-332), please quote all your uncertainties in the text, diagrams and tables at the 2 sigma level.

We choose not to use “sigma” notation for our age uncertainties, but instead use sd (standard deviation). Since there is nothing “magical” about 2 sd or 2.5 sd or 3 sd, we prefer to report our age uncertainties in terms of 1 sd, and this is consistent and clear throughout the manuscript. We have checked the uncertainty in the determination of the age of the Temora2 zircon reference material, and our reported value of 421.1 ± 2.8 Ma is correct, with the uncertainty stated at the 1 sd level.

-p.6 line 189. I agree that no significant bias is present in the U/Pb determinations but the key results of this paper are reported in p.3 as mean $^{207}\text{Pb}/^{206}\text{Pb}$ ages. Consequently, if the inter-element fractionation is well corrected, you could also provide a few values demonstrating that $^{207}\text{Pb}/^{206}\text{Pb}$ mass bias is also well defined.

We feel that discussion of the mass bias in $^{207}\text{Pb}/^{206}\text{Pb}$ is beyond the scope of this paper.

-p.16 Extended Data Table 1: I am not really happy with that table (see again Horstwood et al., 2016). (1) I am not sure that the primary current is useful to the reader. (2) Since Th and U contents are reported, an additional column for the Th/U ratios would be interesting. (3) $^{206}\text{Pb}/^{238}\text{U}$ and $^{207}\text{Pb}/^{206}\text{Pb}$ ratios and corresponding uncertainties (preferably 2 s) are required. (4) $^{206}\text{Pb}/^{238}\text{U}$ and $^{207}\text{Pb}/^{206}\text{Pb}$ apparent ages and corresponding uncertainties at the 2 s level are required. (5) Considering the range of ages and errors, digits after the point are useless for Archean zircons. (6) Due to the large errors for old zircons and the very long ^{232}Th decay constant for the young ones, $^{206}\text{Pb}/^{232}\text{Th}$ apparent ages are not necessary.

We feel that the primary current used in the SIMS determinations is indeed useful, because it illustrates the differences in analytical conditions we used for the Archean vs. Miocene zircons. We have added a column for Th/U to Supplementary Table 1 (see above). We have changed the number of significant figures on all ages and uncertainties (1 sd level) reported in Supplementary Table 1.

Jean-Louis Paquette